# The development and validation of a social media fatigue scale: From a cognitive-behavioral-emotional perspective

**Shiyi Zhang[1], Yanni Shen[1], Tao Xin[1]\*, Haoqi Sun[1], Yilu Wang[2], Xiaotong Zhang[3], Siheng Ren[4]**

**1** Collaborative Innovation Center of Assessment for Basic Education Quality, Beijing Normal University, Beijing, China, **2** School of Psychological and Cognitive Sciences, Peking University, Beijing, China, **3** King's College London, London, United Kingdom, **4** Space Engineering University, Beijing, China

\* xintao@bnu.edu.cn

**Data Availability Statement:** All relevant data are within the manuscript and its Supporting information files.

## Abstract

Social media fatigue (SMF), which refers to social media users' tendency to withdraw from social media because of feeling overwhelmed, is closely related to individuals' social life and well-being. Many studies focused on understanding SMF and exploring its enablers and influences. However, few pieces of research administered a standard measurement of SMF. This study aimed to develop and validate a measure of SMF, and a cross-sectional survey was conducted among 1599 participants in total. Semi-structured interviews of 30 participants were firstly conducted as a pilot study, and an initial version of the social media fatigue scale (SMFS) with 24 items was generated. Then, both exploratory factor analysis (N = 509) and confirmatory factor analysis (N = 552) as well as reliability and validity analysis (N = 508) were conducted and a 15-item SMFS was finally developed. The results demonstrated that: 1) SMF was a multi-dimension concept including a cognitive aspect, an emotional aspect and a behavioral aspect; 2) the three-dimensional structure of the SMFS (cognitive-behavioral-emotional structure) fitted the data well; 3) the McDonald's Omega coefficients for the SMFS was 0.83, suggesting that the SMFS was reliable; 4) criterion validity was satisfactory as indicated by both the significant correlations between self-rated scores of fatigue and total SMFS scores and the significant regression model of SMF on social media privacy, social media confidence, and negative feeling after comparison. Based on the Limited Capacity Model, the present study expanded SMF from a unidimensional model to a three-dimension model, and developed a 15-item SMFS. The study enriched the existing knowledge of SMF, and coined a reliable and valid tool for measuring it. Besides, concluding the typical characteristics of SMF, the study may provide some inspiration for both researchers and social media managers and operators in mitigating SMF.

## Introduction

Social media shorten the space distance between people and provide an online platform for them to make friends and to post information about themselves. By the end of 2018, the number of social media users in the world had exceeded 3.48 billion, about four times as many as

**Funding:** This work was supported by a research grant from the Major Projects of National Social Science Foundation of China (grant number 19ZDA359) to TX. The funder had no role in study design, data collection and analysis, decision to publish, or preparation of the manuscript.

**Competing interests:** The authors have declared that no competing interests exist.

that in 2010, accounting for 45 percent of the global population. Several surveys, however, showed that individuals tend to engage themselves less in social media in spite of the increasing popularity of social network services [1, 2]. For example, Facebook users were once found to update their online status or share life experiences less frequently than before [3]. Similar phenomenon also appears in China, according to the 42rd and 43rd *China Statistical Report on Internet Development*, the use of WeChat and Q-Zone (Chinese mainstream social media applications) declined from 87.3% and 64.4% to 83.4% and 58.8% respectively [4, 5]. Such a phenomenon has motivated researchers to explore the underlying causes, and during the process of exploration, a new concept named social media fatigue (SMF) was proposed to explain the discontinuous social media use.

## Social media fatigue (SMF)

The generally accepted definition of SMF refers to user's tendency to withdraw from social media usage because of feeling overwhelmed by social media [6]. Such definition indicates that SMF is a kind of subjective feeling, and Bright et al. once utilized such definition in their research to reveal a typical characteristic of SMF from cognitive perspective, which is cognitive overload [3]. Except for that, SMF has other characteristics in emotional and behavioral aspects. For instance, some researchers described SMF as afflictions that people have when using social media [7, 8]. These afflictions could be tiredness, boredom, burnout, indifference or lower interest [9], which can be concluded as emotional features of SMF. As for behavioral features, however, except for the decreasing frequency of social media use, researchers focused on very few symptoms. Ream and Richardson once analyzed the concept of fatigue and pointed out that fatigue could come with affective, behavioral and cognitive responses, including dissipation of attention, forgetfulness, inability to make decisions, inability to cope, and many other symptoms [10]. Similarly, Huff once described three behavioral signs of SMF, including procrastination, lack of focus and temporary amnesia [11]. As SMF is a kind of fatigue toward social media, it is reasonable to believe that SMF may have other behavioral characteristics besides the decreasing frequency of social media use, like forgetfulness and lack of focus.

Except for typical signs of SMF, researchers are also interested in what kind of factors may lead to SMF. Bright et al. found that users' subjective feelings of social media, including social media confidence and privacy concerns, could predict users' SMF significantly [3]. Social media confidence refers to the extent to which users believe they can use social media in an effective manner, and the more confidence users have, the less social media fatigue they will feel. Privacy concern refers to users' concerns that using social media may leak out their private information. Quite reverse to social media confidence, privacy concerns have a negative predictive effect on SMF. To be specific, the more concerned the users are with privacy disclosure, the more social media fatigue they will feel. Lim and Choi once verified such effect, apart from which, they also found that social comparison could accentuate users' emotional exhaustion [12]. For further exploration, Chinese researchers pointed out that it was negative feeling after comparison that aggravated the level of SMF [13].

## Theoretical backgrounds of SMF

The Limited Capacity Model (LCM) has been utilized to describe the features of SMF and to explain how SMF happens. LCM assumes that people are information processors and they require processing resources to encode, store, and ultimately retrieve the information they receive [14]. The processing resource, however, are limited in amount. When faced with too much information, human brains will become overloaded, making the processing resources

insufficiently available. Similarly, when exposed to long-term excessive information, individuals with SMF will feel overloaded and eventually tend to refrain from using social media. Bright et al. once utilized LCM to reveal characteristics of SMF from a cognitive perspective. In fact, LCM can also be used to explain characteristics of SMF in other aspects, like emotional and behavioral aspects.

First, SMF is usually accompanied with receiving excess information. According to LCM, long term exposure to excessive information makes individuals with SMF feel overloaded and ultimately tend to reduce their use of social media. That is, overloaded information in cognition is an important part of SMF. Second, behavioral symptoms like forgetfulness and a lack of focus, are another key component of SMF. Limited capacity model emphasized that overwhelming information could affect memory, and studies have showed that individuals feel difficult to concentrate on a task or even quite distinctive things in tasks demanding extensive cognitive loading [15, 16]. In addition, faced with information overload, people with SMF cannot find useful material to update their posts and lack cues to remind them of their original intentions [17, 18]. Third, emotional reaction should be included in SMF. People are more prone to suffering from negative emotions, such as frustration and anxiety, when faced with information overload [19, 20]. It can also prevent people from adequately regulating and controlling their moods [21–23]. Negative emotional arousal can thus be regarded as a part of SMF.

## Measures of SMF

Researchers usually developed or revised different items for their own research purposes. The accepted method of measuring SMF was first described by Bright et al., who devised five items rated on a seven-point scale adapted from a study of Gartner [3]. This SMF questionnaire mainly measures the feelings of information overload, and is capable of good reliability (0.91) with no test for validity [3]. Based on the study of Bright et al., Dhir et al. used some of the items to measure SMF [2]. In these studies, SMF was measured from the view of cognitive overload.

Researchers also measure SMF from the view of emotional experiences. Lin combined four items from two researches to assess SMF, which focuses on diminished attraction and novelty, the decreasing frequency of social media use, and diminished satisfaction [8, 24, 25]. The psychological properties of this measure were not reported. In addition, Cramer, Song, and Drent devised three items rated on a five-point scale to measure Facebook fatigue with one item measuring negative feelings and two items measuring the motivation to use Facebook [26]. This study also only presented the reliability (0.77) of the SMF measure.

For the SMF of people in mainland China, Zhang, Li, and Peng utilized two items in Bright's study to measure SMF, the reliability of which was 0.73 [27]. Zhang, Zhao, Lu and Yang employed individuals' attitudes and negative experiences, and finally developed six items to measure social network fatigue with a reliability of 0.92 [9]. This social network fatigue measure, as a part of the research instruments, was adapted from previous studies and followed a standard process to be revised in Chinses version. However, it mainly focuses on testing the emotional component to a certain situation in Q-zone (a kind of social media in mainland China).

In short, although the existing measures of SMF possess good reliability, other psychological properties of those measures are still unknown, and what's more, uniform measurements of SMF are still lacking. On the other hand, studies mainly focused on a certain situation, such as Facebook or Q-zone, thus making the measurement of SMF incomprehensive.

## The present study

Many studies focus on understanding SMF and exploring its relationships with psychological determinations, however, to date few researches have used a standard psychological

measurement of SMF. To fill this gap, the present study aimed to develop and validate a measurement of SMF.

In order to coined items, a pilot study was first conducted. In the pilot study, we designed several questions and had semi-structured interviews with 30 people. Based on the results we concluded specific symptoms of SMF and coined a twenty-four-item social media fatigue scale (raw scale). Then we invited those people again to complete the raw scale and provide suggestions of improving the wordings of the items, and the preliminary scale contained 24 items was thus developed.

Then the exploration and confirmation of the scale's structure were conducted successively, and we got the final scale, consisting of 15 items. In order to evaluate the psychometric properties of the final scale, we computed McDonald's omega ($\omega_t$ and $\omega$s) of both the scale and subscales to evaluate the reliability of the scale. We also collected concurrent criterion and other variables related to SMF to assess the validity of the scale. Specific details were further explained in both method part and result part.

## Methods

### Sample and procedure

The study was approved by the ethics review committee of the Faculty of Psychology in Beijing Normal University. As for the pilot study, 30 participants (22 women, mean age = 22.27) were recruited offline for semi-structured interviews using convenience sampling (Fig 1). For scale development and validation, simple random sampling was conducted among subject databases of SO JUMP (an online platform providing sample collection services for a fee) and three groups of participants were recruited online successively. The inclusion criterion was that every participant had used social media within one month before he/she took part in the study. Besides, according to response time and other factors, the platform could automatically eliminate those participants who didn't complete the scale seriously. Thus, the final valid sample size for scale development and validation was 1569. Data of 509 participants (Sample 1: 311 women, mean age = 24.75) aged between 18 and 39 years were used to explore the scale's structure; data of 552 participants (Sample 2: 330 women, mean age = 28.75) aged between 18 and 39 years were used to verify the scale's structure and evaluate its properties; data of 508 participants (Sample 3: 344 women, mean age = 26.83) aged between 18 and 35 years were used to evaluate measuring properties of the scale. Descriptive statistics of three samples were shown in Table 1. Informed consents were obtained from all participants.

### Measures

**Social media fatigue scale (SMFS).** The preliminary version of the scale consisted of 24 items. Each item was ranked on a seven-point scale (1 = *totally disagree*, 7 = *totally agree*).

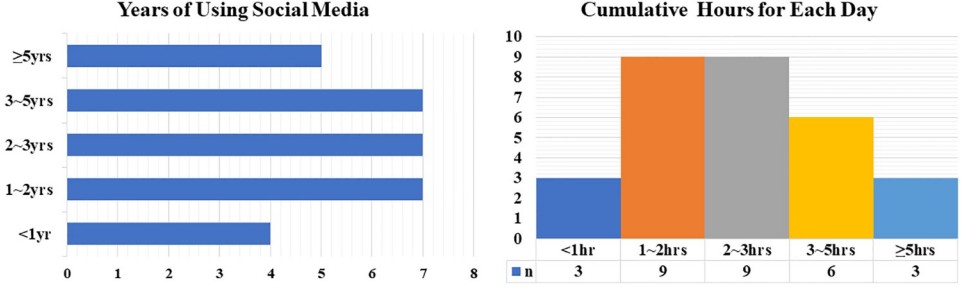

**Fig 1.**

**Table 1. Descriptive statistics of Sample 1–3.**

| Character | Category | Sample 1 | | Sample 2 | | Sample 3 | |
|---|---|---|---|---|---|---|---|
| | | N | Percentage | N | Percentage | N | Percentage |
| Sex | Male | 197 | 38.70% | 222 | 40.22% | 164 | 32.28% |
| | Female | 311 | 61.10% | 330 | 59.78% | 344 | 67.72% |
| | Missing | 1 | 0.20% | 16 | 2.90% | 60 | 11.81% |
| Age | 18–20 | 47 | 9.23% | 98 | 17.75% | 142 | 27.95% |
| | 21–25 | 287 | 56.39% | 233 | 42.21% | 168 | 33.07% |
| | 26–30 | 122 | 23.97% | 203 | 36.78% | 138 | 27.17% |
| | 31–35 | 44 | 8.64% | 2 | 0.36% | 0 | 0% |
| | 36–40 | 9 | 1.77% | 0 | 0% | 0 | 0% |
| Education Level | Less than high school | 0 | 0% | 0 | 0% | 2 | 0.39% |
| | High school | 2 | 0.39% | 11 | 1.99% | 5 | 0.98% |
| | Junior college | 8 | 1.57% | 59 | 10.69% | 47 | 9.25% |
| | Undergraduate | 11 | 2.16% | 434 | 78.62% | 405 | 79.72% |
| | Master | 326 | 64.05% | 44 | 7.97% | 46 | 9.06% |
| | PhD | 137 | 26.92% | 4 | 0.72% | 3 | 0.59% |

Finally, 9 items were excluded and the final version of SMFS comprised 15 items with a McDonald's Omega coefficient of 0.83 in this study.

**Psychological and demographic variables.** Participants' self-rated scores (SRS, "Are you tired of using social media?") for SMF were collected as a concurrent criterion. According to Bright et al., participants' privacy could predict their levels of SMF positively, and social media confidence could predict their levels of SMF negatively [3]. In order to evaluate the validity of the developed scale, participants' privacy concerns and confidence regarding social media were also measured. Demographic variables such as sex, age, and education level were also collected.

## Data analyses

To develop a reliable and valid SMFS, the present study utilized several analysis methods including Exploratory Factor Analysis (EFA), Confirmatory Factor Analysis (CFA), reliability and validity assessment.

Firstly, a series of EFAs with oblique GEOMIN rotation was used to both explore the structure of SMFS and delete items (using Sample 1). According to the eigenvalues of each factor, one to four factors were extracted successively (eigenvalues >1). The ideal model was chosen according to goodness-of-fit indices, including Comparative fit index (CFI), Tucker-Lewis index (TLI), root mean square error of approximation (RMSEA) and standardized root mean square residual (SRMR). The following criteria were used to evaluate which model was acceptable: An RMSEA value less than 0.08 indicates an acceptable fit for model, and value less than 0.05 indicates a good fit for model [28]. A TLI value more than 0.90 indicates an acceptable fit [29]. A CFI value more than 0.95 combined with a SRMR value less than 0.08 also correspond to an acceptable fit [30].

During the process of item deleting, EFA was conducted for several times and only one item could be deleted at each time. Whether the item should be deleted was based on its loading, and the item with loading value less than 0.4 or having cross-loadings was supposed to be deleted [31]. If there were more than one item meeting with such deletion standard at one

time, the content of them would be taken into account. The analysis would not end until there was no item meeting with deletion standard and the goodness-of-fit indices were acceptable.

Secondly, CFA was generated to verify the structure of SMFS (using Sample 2). During such an analysis, the fit indices and factor loadings were calculated. Good fit indices and significant factor loadings indicate that the final version of SMFS derived from EFA was acceptable.

Thirdly, to evaluate the properties of SMFS, the present study recruited another set of participants (Sample 3), and assessed the reliability and validity of the 15-item SMFS. As for reliability, McDonald's Omega coefficients were computed for the scale and subscales. As for validity, concurrent validity and construct validity were tested.

The statistical analyses were completed using MPLUS 7.11, SPSS 18.0 and R program (psych package). Missing data (only in Sample 1) were not imputed considering both the low missing rate (2.6% cases, 13 cases) and the Little's MCAR analysis result ($\chi^2 = 233.48$, $df = 230$, $p > 0.05$).

## Results

### Typical symptoms of SMF

In pilot study, we sorted out and summarized participants' answers and concluded the typical symptoms of SMF in cognitive, behavioral and emotional aspects respectively. Based on these symptoms, 24 items (raw items) were then generated.

As for the cognitive aspect, we chose cognitive overload as a typical manifestation, which was accordant to the description in Bright's study [3]. Taking both previous studies and the results of interviews into account, we coined eight initial items, five of which were adapted from Bright's study [3], the other three were newly generated.

As for the behavioral aspect, people reported that they felt less motivated to recommend their social media accounts to others, and that they always forgot what they had browsed or had intended to browse on social media. Based on the reports, we concluded both delaying use of social media and being forgetful when using social media as typical characteristics, and coined eight items to describe these behaviors.

As for the emotional aspect, we found that people with high level of SMF were prone to getting exhausted, wary, anxious and irritated when using social media, which was in accordance with previous researches in mainland China [9, 32]. According to both the reports and previous researches, we generated eight items for emotional dimension.

After the generation of the raw items, the 30 participants were re-invited to complete the raw scale and put forward suggestions on improving the wording of the item. According to their feedback, the preliminary scale was developed (Table 2).

### Item analysis of SMFS

The preliminary scale included 24 items. The mean scores, skewness, kurtosis and corrected item-total correlations (CITC) of each item for Sample 1 were showed in Table 2. The Kurtosis index indicated that both samples' responses deviated from a normal distribution, and the weighted least square mean and variance method was therefore used for both EFA and CFA [33]. Items with a low CITC could be considered as problematic items. Thus, with CITC less than 0.3 as the standard [34], we excluded item 8, item 13 and item 16 from subsequent analysis.

### Exploratory factor analysis of SMFS

The EFA assumptions were first tested using Sample 1. The Kaiser-Meyer-Olkin measure *KMO* equaled 0.92, and Bartlett's test was significant ($\chi^2 = 3325.04$, $df = 210$, $p < 0.001$).

**Table 2. Mean scores, standard deviation, measures of distribution and the corrected item-total correlation for the 24-item social media fatigue scale based on Sample 1.**

| Item content | Mean | SD | Skewness | Kurtosis | CITC |
|---|---|---|---|---|---|
| 1 I find that social media sites do not have enough detail to quickly find the information I am looking for. | 4.00 | 1.44 | -0.13 | -0.66 | 0.35 |
| 2 I am frequently overwhelmed by the amount of information available on social media sites. | 3.80 | 1.64 | 0.10 | -0.84 | 0.58 |
| 3 When searching for information on social media sites, I frequently just give up because there is too much to deal with. | 3.92 | 1.69 | -0.14 | -0.97 | 0.52 |
| 4 I am likely to receive too much information when I am searching for something on social media sites. | 4.92 | 1.38 | -0.55 | -0.06 | 0.33 |
| 5 The amount of information available on social media sites makes me feel tense and overwhelmed. | 3.48 | 1.62 | 0.20 | -0.87 | 0.62 |
| 6 I usually avoid using social media for having received too much information. | 4.38 | 1.76 | -0.27 | -0.96 | 0.44 |
| 7 I feel angry when I realize that social media has taken up too much of my time. | 4.58 | 1.69 | -0.37 | -0.82 | 0.40 |
| 8 After watching the short-form videos and reading the snippets posted by others, I'm not willing to repost them. | 5.22 | 1.41 | -0.75 | 0.08 | 0.14 |
| 9 I always have no idea what I am going to post on social media. | 4.07 | 1.76 | -0.07 | -0.98 | 0.49 |
| 10 When I login a social media site, I'll always forget whom I've intended to stalk on the site. | 3.53 | 1.66 | 0.22 | -0.88 | 0.52 |
| 11 I'm likely to forget the content of the status which I have intended to repost. | 4.06 | 1.67 | -0.06 | -0.90 | 0.45 |
| 12 When I open a social media site, I may forget what I've intended to post on the social media site. | 3.60 | 1.66 | 0.20 | -0.89 | 0.56 |
| 13 I would rather do something else than update my posts on social networks. | 4.36 | 1.59 | -0.12 | -0.77 | 0.27 |
| 14 It's hard for me to come up with good ideas for updating status on social media sites. | 4.16 | 1.71 | -0.22 | -0.95 | 0.47 |
| 15 I usually forget what I've read on the social media quickly. | 4.29 | 1.62 | -0.15 | -0.77 | 0.35 |
| 16 Sometimes I hardly realized that I should have posted what I've eaten on my social media until I finished eating. | 4.06 | 1.92 | -0.14 | -1.11 | 0.24 |
| 17 I feel annoyed when I find there is too much unread information on social media sites. | 3.21 | 1.60 | 0.50 | -0.54 | 0.54 |
| 18 Functions in the social network (check-in, status updates, etc.) make me irritated. | 3.65 | 1.83 | 0.16 | -1.14 | 0.63 |
| 19 I feel anxious when I was referred to (@) by others on the social media sites. | 2.87 | 1.63 | 0.77 | -0.27 | 0.60 |
| 20 I feel nervous when receiving friend requests on social media sites. | 3.37 | 1.68 | 0.34 | -0.87 | 0.54 |
| 21 Using several social media platforms always makes me feel at a loss. | 3.65 | 1.76 | 0.19 | -1.06 | 0.59 |
| 22 Before I login in my social media account, I usually fear of receiving too much new messages. | 3.49 | 1.73 | 0.26 | -0.96 | 0.57 |
| 23 I feel angry seeing people always talking about the details of their life on social media. | 3.33 | 1.64 | 0.26 | -0.92 | 0.54 |
| 24 I will constantly browse the social media (such as Weibo, WeChat, etc.) for fear that I have missed important information. | 3.96 | 1.80 | -0.15 | -1.09 | 0.31 |

The anti-image correlation indices were all above 0.5, indicating that the data satisfied the assumptions.

EFA was then conducted using Sample 1, and 1 to 4 factors were extracted. The scree plot is presented in Fig 2, and Table 3 provides fit indices for each model, and Table 4 shows the results of model comparison; three-, and four-factor models seemed to fit the data well (*CFIs* >0.95, *TLIs* >0.90, *RMSEAs* <0.08, *SRMRs* <0.08). However, based on the literature review and structure assumption for SMF, the factor loadings pattern for the current three-factor model was nearly the assumed structure for SMF and it was thus chosen as the optimal SMFS structure. The factors were named cognitive experiences (CE), behavioral experiences (BE), and emotional experiences (EE), and the eigenvalues for the sample correlation matrix were 5.47, 1.43, and 1.13, respectively.

Although a three-factor model was chosen, item 1, item 24, item 15, item 23 had low loadings (loadings<0.4), and item 5 had a high loading (0.61) on an unexpected factor. Therefore, these unsatisfactory items were deleted one by one, and EFA was conducted at each deletion step. Moreover, in order to make sure that numbers of items within each factor were the same, we deleted item 21 which had the lowest item loading in its factor (0.45), and a 15-item SMFS was ultimately established (*CFI* = 0.96, *TLI* = 0.93, *RMSEA* = 0.08, *SRMR* = 0.04). Factor loadings for the 15-item scale are shown in Table 5. All items had acceptable loadings (>0.4) on only one factor.

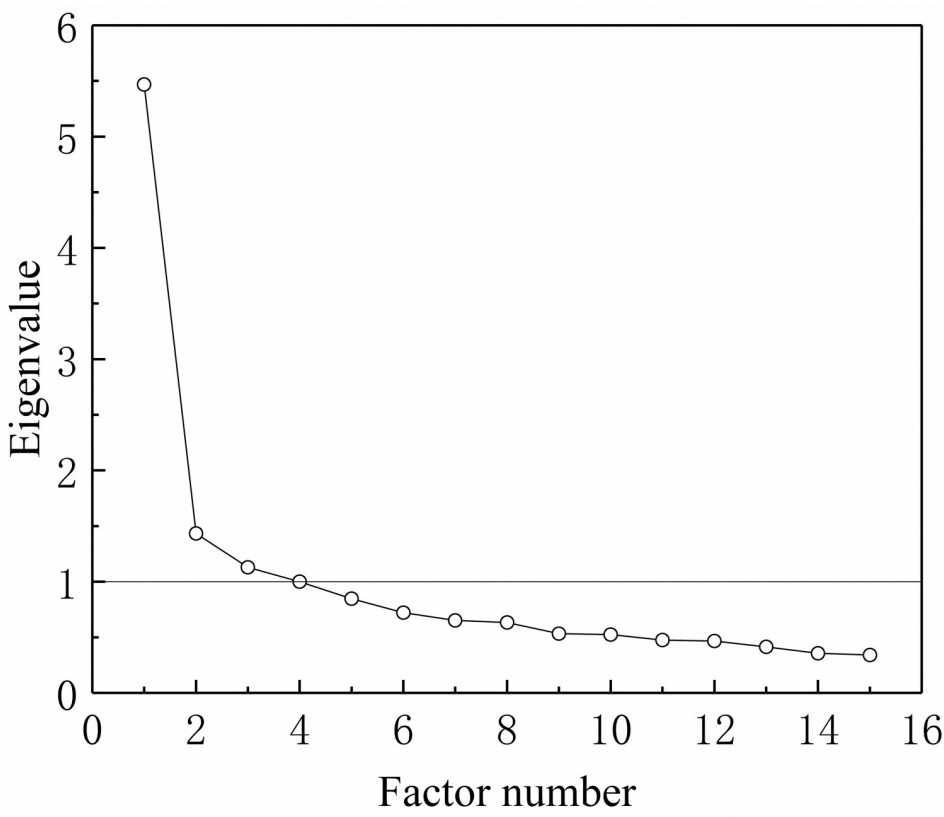

**Fig 2.**

## Confirmatory factor analysis of SMFS

CFA was employed to test the three-factor structure of the 15-item SMFS derived from EFA. Based on Sample 2, the fit indices indicated that three-factor model fit the data well ($CFI$ = 0.97, $TLI$ = 0.96, $RMSEA$ = 0.06). Standardized factor loadings, as shown in Fig 3, were all significant, ranging from 0.43 to 0.73. Standardized intercorrelation coefficients between factors ranged from 0.66 to 0.83.

## Reliability and validity of SMFS

To evaluate the 15-item SMFS for reliability, McDonald's Omega coefficients were computed for the scale and subscales. The 15-item SMFS had a satisfactory reliability of 0.83, exceeding

**Table 3. Summary of model fit indices for EFA.**

| n of factor | SRMR | Chi-Square | df | CFI | TLI | RMSEA (90% CI) |
|---|---|---|---|---|---|---|
| 1 | 0.07 | 1056.34 | 189 | 0.89 | 0.87 | 0.10 (0.09, 0.10) |
| 2 | 0.05 | 638.32 | 169 | 0.94 | 0.92 | 0.07 (0.07, 0.08) |
| 3 | 0.04 | 493.47 | 150 | 0.95 | 0.94 | 0.07 (0.06, 0.07) |
| 4 | 0.03 | 382.59 | 132 | 0.97 | 0.95 | 0.06 (0.05, 0.07) |

Note: The statistics shown above were based on the initial SMFS (24-item SMFS). SRMR = Standardized Root Mean Square Residual, CFI = Comparative Fit Index, TLI = Tucker-Lewis Index, RMSEA = Root Mean Square Error of Approximation;

**Table 4. Summary of model comparisons.**

| Models Compared | Chi-Square | df | P-Value |
|---|---|---|---|
| 1-factor against 2-factor | 296.29 | 14 | 0.000 |
| 2-factor against 3-factor | 113.97 | 13 | 0.000 |
| 3-factor against 4-factor | 84.39 | 12 | 0.000 |
| 4-factor against 5-factor | 82.17 | 11 | 0.000 |

Note: The model comparisons were based on Sample 1 (n = 509, age: 18–39).

**Table 5. EFA factor loadings for the three-factor model of the 15-item SMFS.**

| Item content | Mean (SD) | f1 | f2 | f3 |
|---|---|---|---|---|
| 2 I am frequently overwhelmed by the amount of information available on social media sites. | 3.80 (1.64) | **0.63**\* | 0.12 | 0 |
| 3 When searching for information on social media sites, I frequently just give up because there is too much to deal with. | 3.92 (1.69) | **0.50**\* | 0.16\* | 0 |
| 4 I am likely to receive too much information when I am searching for something on social media sites. | 4.92 (1.38) | **0.56**\* | -0.1 | -0.17\* |
| 6 I usually avoid using social media for having received too much information. | 4.38 (1.76) | **0.43**\* | 0.02 | 0.08 |
| 7 I feel angry when I realize that social media has taken up too much of my time. | 4.58 (1.69) | **0.47**\* | 0.04 | 0.01 |
| 9 I always have no idea what I am going to post on social media. | 4.07 (1.76) | 0.09 | **0.60**\* | -0.04 |
| 10 When I login a social media site, I'll always forget whom I've intended to stalk on the site. | 3.53 (1.66) | 0.29\* | **0.40**\* | 0.04 |
| 11 I'm likely to forget the content of the status which I have intended to repost. | 4.06 (1.67) | 0.12 | **0.51**\* | -0.02 |
| 12 When I open a social media site, I may forget what I've intended to post on the social media site. | 3.60 (1.66) | 0.01 | **0.62**\* | 0.16\* |
| 14 It's hard for me to come up with good ideas for updating status on social media sites. | 4.16 (1.71) | -0.04 | **0.63**\* | 0.03 |
| 17 I feel annoyed when I find there is too much unread information on social media sites. | 3.21 (1.60) | 0.12 | -0.09 | **0.69**\* |
| 18 Functions in the social network (check-in, status updates, etc.) make me irritated. | 3.65 (1.83) | 0.35\* | 0 | **0.47**\* |
| 19 I feel anxious when I was referred to (@) by others on the social media sites. | 2.87 (1.63) | -0.07 | 0.12\* | **0.74**\* |
| 20 I feel nervous when receiving friend requests on social media sites. | 3.37 (1.68) | 0.01 | 0 | **0.72**\* |
| 22 Before I login in my social media account, I usually fear of receiving too much new messages. | 3.49 (1.73) | 0.05 | 0.01 | **0.72**\* |
| Cognitive Experiences (*f1*) | | - | | |
| Behavioral Experiences (*f2*) | | 0.59\* | - | |
| Emotional Experiences (*f3*) | | 0.56\* | 0.54\* | - |

Note:

\* = p <.05. The statistics above were based on Sample 1 (n = 509, age: 18–39). Item 1, 24, 15, 23, 5 and 21 were deleted successively. Statistics in the last three rows show intercorrelations between three factors.

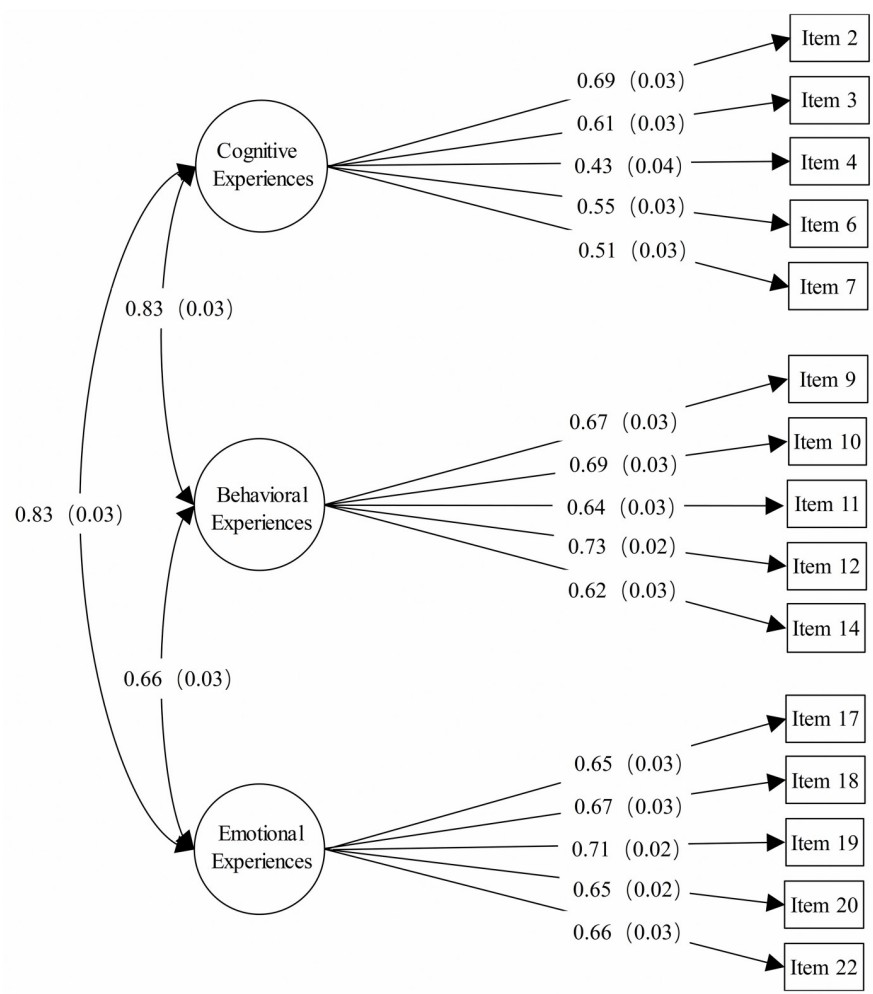

**Fig 3.**

the minimum of 0.80 [35, 36]. As shown in Table 6, the McDonald's Omega coefficients for subscales were 0.61, 0.76, 0.71 respectively and the correlation coefficients varied between 0.46 and 0.50 indicating that the reliability of each factor was acceptable and the three factors were adequately distinct [37].

SRS was collected as a concurrent criterion. SRS had significantly positive correlations with both total score (TS, $r_{SRS-TS} = 0.51$, $p < 0.001$) and subscale scores ($r_{SRS-cog} = 0.47$, $r_{SRS-behav} =$

**Table 6. Reliability of subscales and correlations between each dimension.**

| Subscales | $\omega_s$ | 1 | 2 | 3 |
|---|---|---|---|---|
| 1 Cognitive Experiences | 0.61 | 1 | | |
| 2 Behavioral Experiences | 0.76 | 0.50 | 1 | |
| 3 Emotional Experiences | 0.71 | 0.50 | 0.46 | 1 |

Note: Correlation coefficients were computed by sum scores of each dimension, and all of the coefficients were significant at 5% level (two-tailed).

0.35, $r_{SRS\text{-}emo}$ = 0.43, $p$ <0.001). To further validate the 15-item SMFS, participants with different SRS were divided into three groups (low, medium and high group), and one-way analysis of variance (ANOVA) was used to test whether their TSs were significantly different. Homogeneity of group variances was first tested using Levene's test, and the results (Table 7) indicated that the data satisfied the homogeneity of variances assumption. The group difference in SRS mean scores was significant at 0.001 ($F_{(2,\ 505)}$ = 66.37). To verify that the SRS order was reasonable, a least significant difference test was performed, and all comparisons were found to be significant. Moreover, higher SRS values corresponded to higher TS values. This demonstrated that the order was reasonable as expected.

Bright et al. supposed and verified that privacy concerns could positively predict participants' degrees of SMF, whereas social media confidence could negatively predict it [3]. Besides, Niu found that negative feelings after comparison aggravated the level of SMF [13]. Thus a regression model with privacy concern, social media confidence and negative feelings after comparison as predictors and TS as a dependent variable was then constructed to evaluate criterion-related validity. Results showed that social media confidence ($B$ = -1.62, $p$ <0.001), privacy concern ($B$ = 1.17, $p$ <0.001) and negative feelings after comparison ($B$ = 0.615, $p$ <0.001) were significantly related to SMF (Table 8). In addition, the CFA results also indicated that the 15-item SMFS had a satisfactory construct validity ($CFI$ = 0.97, $TLI$ = 0.96, $RMSEA$ = 0.06).

## Discussion

The present study is the first study to our knowledge to develop and validate an SMF measurement for mainland Chinese. In general, the present study adopted semi-structured interviews to preliminary determine the three-factor structure of SMF and coined twenty-four items to form an initial SMF scale. Then the study explored the structure of the scale by conducting EFA, and deleted items with low loadings. When the three-factor structure had become clear

**Table 7. Descriptive statistics for the self-rated SMF score and the test of homogeneity of variance.**

| Self-rated Score | M | N | SD | Levene's statistic | df1 | df2 | Sig. |
|---|---|---|---|---|---|---|---|
| 1 | 46.15 | 147 | 11.71 | 0.174 | 2 | 505 | 0.840 |
| 2 | 57.47 | 331 | 11.77 | | | | |
| 3 | 67.57 | 30 | 11.32 | | | | |

Note: M = mean, N = number of participants, SD = Standard Deviation, df = degree of freedom, Sig. = significance level.

**Table 8. Regression analysis result.**

| | Unstandardized estimates | Standard error | Standardized estimates | t | p-level | Collinearity statistics | |
|---|---|---|---|---|---|---|---|
| | B | SE | β | | p | Tol | VIF |
| Intercept | 52.19 | 4.41 | | 11.84 | 0.00 | | |
| Social media privacy | 1.17 | 0.15 | 0.30 | 7.90 | 0.00 | 0.97 | 1.04 |
| Social media confidence | -1.62 | 0.19 | -0.32 | -8.34 | 0.00 | 0.99 | 1.01 |
| Negative feeling after comparison | 0.62 | 0.11 | 0.22 | 5.77 | 0.00 | 0.97 | 1.04 |
| $F_{(3,\ 504)}$ | 67.69 | | | | | | |
| $R^2$ | 0.29 | | | | | | |
| Adjusted $R^2$ | 0.28 | | | | | | |

Note: Social media privacy, social media confidence and negative feeling after comparison were predictors, and social media fatigue was dependent variable.

and stable, CFA was then utilized to verify such a structure. Then the McDonald's Omega coefficients, criterion validity and construct validity were examined to evaluate the properties of the 15-item SMF scale. Based on a cognitive-behavioral-emotional framework, the 15-item SMFS with acceptable reliability and validity was finally developed.

One major contribution of this study is that it clarified the three-factor structure of SMF. As we mentioned in former part of this article, although a great many researchers focused on SMF and tried to make it clear what leads to SMF and how SMF influences psychological characteristics of human beings. Most researchers only regarded SMF as a concept with single dimension. For instance, Bright et al. once utilized LCM as a framework to examine the concept of SMF [3]. The explanation was quite convincing, however, it only focused on the cognitive characteristics of SMF. Taking another study as an example, Lee, Son and Kim once explored how social media characteristics affect user's SMF level. In their study, however, SMF was defined as a subjective feeling of tiredness in social media use [38]. The two studies defined SMF from quite different perspectives, according to which we cannot help asking that what SMF indeed is?

In order to answer the question, we conducted several semi-structured interviews and asked participants to describe their actual experiences and feelings about social media fatigue. The results showed that apart from feeling information overloaded, people with high levels of SMF always had low motivations of using social media and always forgot what they'd browsed on social media. Besides, they also reported that they felt anxious and irritated. Thus, we concluded that SMF was a multi-dimension concept including a cognitive aspect, an emotional aspect and a behavioral aspect.

The second major contribution of this study is that the present study expanded the utilization of LCM to interpret SMF, from only one perspective (cognitive aspect) to three perspectives (cognitive, behavioral and emotional aspects). For the cognitive dimension, LCM theory explains that immersion in information consumes a great many mental resources and can make people fall into a state of overload easily. Trapped in such a state for a long time, people tend to feel exhausted and fatigue. For the behavioral dimension, cognitive overload requires more mental resources to be allocated to processing information so that mental resources for memory retrieval would become deficient, resulting in temporary amnesia and poor prospective memory. For the emotional dimension, mental resources for mood control will also become deficient because of information overload, and users with SMF will therefore experience negative emotions. The present study expanded LCM from only explaining SMF by cognitive overload to elaborating behavioral and emotional characteristics of SMF. To some extent the present study enriched the application of LCM.

The third major contribution of this study is that it has developed an SMFS in a relative reliable way, and makes up for the lacks of previous studies which only used few items (usually, those items varied in different studies) to assess SMF without detailed reports of their psychometric properties. The present study used semi-structured interview and data analysis to explore the structure of SMF and finally coined an SMFS. Furthermore, both reliability and validity assessment were conducted. In the reliability test, the McDonald's Omega coefficients indicated that the 15-item SMFS had an acceptable reliability. The significantly positive correlation between TS and SRSs, significant one-way ANOVA results, and the reproduction of the regression model indicated that the 15-item SMFS was a valid measurement scale. In this way, the present study provided researchers with an effective diagnostic tool of assessing SMF.

In addition, the fourth major contribution is that the present study showed main characteristics of SMF, which may also provide references for researchers, social media managers and operators to look for ways of mitigating people's SMF. Firstly, the study verified that it was cognitive overload that served as one of the main characteristics of SMF. The more information people got accessed to on social media, the more likely they would feel fatigue. In this way,

in order to reduce users' fatigue, operators should provide users with fewer recommendations that are more personalized and can effectively arouse users' interest. Moreover, social media interfaces should also be as much as simplified to mitigate users' cognitive loading. Furthermore, people with SMF usually experience negative emotions, thus tracking users' emotional experiences and setting up a more relaxing online environment is another method to reduce SMF. As for how to figure out users' preferences and make personalized information recommendations, how to set a simple, beautiful and pleasant interface and how to track users' emotions, these topics need further exploration by future researchers.

In conclusion, summarizing the typical performance of SMF including cognitive overload, decreasing motivation of recommending social media accounts to others, delaying use of social media, being forgetful, and being prone to getting negative emotions, and developing a 15-item SMFS, the contribution of this study to the existing research includes both theoretical and practical aspects. Theoretically, the present study summarized the typical characteristics of SMF in cognitive, behavioral and emotional aspects through semi-structured interviews, and explained it with LCM theory. This not only confirmed the understanding of some existing studies on SMF, but also enriched the existing knowledge of SMF and expanded the application of LCM theory in SMF. Besides, the study developed a 15-item SMFS, which verified that SMF does have typical characteristics in cognitive, behavioral and emotional aspects. As for practice aspects, the SMFS developed in the study had good reliability and validity, which enriched the measuring means of SMFS. And in terms of reducing SMF, the research may provide some inspiration for researchers and social media operators.

## Limitations

Considering that the participants should take part in the pilot study twice (semi-structured interviews & revision of the raw scale), we utilized convenient samples in pilot study, but it may lead to the problem of bias. In order to test whether the problem exists, we recruited another small group of participants (N = 32, 16 women, mean age = 30.44) by random sampling method, interviewing them with the same questions as the pilot study, and inviting them to answer the raw scale. The result showed that the consistency existed between the reports of two groups. We also conducted an Independent Samples t Test on the total scores of the raw scale ($M_{pilot}$ = 96.87, $SD_{pilot}$ = 14.41; $M_{new}$ = 104.56, $SD_{new}$ = 27.52) and we found that there was no significant difference between two groups ($t$ = 1.39, $df$ = 47.45, $p$ = 0.17). To some extent, it showed that the problem of bias didn't seem to exist, however, in order to ensure the representativeness of the samples, we earnestly suggest that researchers use random samples in future study.

In terms of other limitations in the present study, it is important to note that we validated a Chinses version of the SMFS, thus sociocultural differences should be considered if researchers from other countries intend to utilize such a measurement directly. In addition, SMFS is a self-reporting scale. The responses are mainly based on participants' subjective feelings and the self-report bias is hardly avoided. To overcome such a problem, SMFS could be used combing with novel methods like analyzing the content of posts on social media and analyzing the participants' network trace. Moreover, the present study wasn't limited to a certain specific social media platform, thus researchers may further revise the SMFS in certain contexts for their own research purposes, such as on Facebook or Twitter.

## Supporting information

**S1 Appendix. Social media fatigue scale (SMFS).**
(DOCX)

**S1 Data. Data of Sample 1.**
(SAV)

**S2 Data. Data of Sample 2.**
(SAV)

**S3 Data. Data of Sample 3.**
(SAV)

## Acknowledgments

The authors would like to express sincere gratitude to all the teachers and classmates who provided help during the research and to all the participants in this study.

## Author Contributions

**Conceptualization:** Shiyi Zhang, Yilu Wang, Xiaotong Zhang.

**Data curation:** Shiyi Zhang, Yilu Wang.

**Formal analysis:** Shiyi Zhang.

**Funding acquisition:** Tao Xin.

**Investigation:** Shiyi Zhang, Yilu Wang, Xiaotong Zhang, Siheng Ren.

**Methodology:** Shiyi Zhang.

**Supervision:** Tao Xin.

**Writing – original draft:** Shiyi Zhang.

**Writing – review & editing:** Yanni Shen, Tao Xin, Haoqi Sun, Siheng Ren.

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
