## [Decision Letter · Decision Letter 0]

21 May 2020

PONE-D-20-08484

The development and validation of a social media fatigue scale: From a cognitive-behavioral-emotional perspective

PLOS ONE

Dear Dr. Xin,

Thank you for submitting your manuscript to PLOS ONE. After careful consideration, we feel that it has merit but does not fully meet PLOS ONE’s publication criteria as it currently stands. Therefore, we invite you to submit a revised version of the manuscript that addresses the points raised during the review process.

We look forward to receiving your revised manuscript.

Kind regards,

Geilson Lima Santana, M.D., Ph.D.

Academic Editor

PLOS ONE

Journal Requirements:

2. PLOS ONE has specific requirements for studies that are presenting a new method or tool as the primary focus, including a newly developed or modified questionnaire or scale (https://journals.plos.org/plosone/s/submission-guidelines#loc-methods-software-databases-and-tools.) One requirement is that the questionnaire or scale must be openly available under a license no more restrictive than CC BY. In light of this, before we proceed, please include a copy of your questionnaire or scale as a Supporting Information file (in the original language) or provide a link if it is available through an online repository. We note that you have already provided an English version of the questions, but we would be grateful if you could also provide the questions in the original language.

Reviewers' comments:

Reviewer's Responses to Questions

**Comments to the Author**

1. Is the manuscript technically sound, and do the data support the conclusions?

Reviewer #1: Partly

Reviewer #2: Yes

2. Has the statistical analysis been performed appropriately and rigorously? 

Reviewer #1: Yes

Reviewer #2: Yes

3. Have the authors made all data underlying the findings in their manuscript fully available?

Reviewer #1: Yes

Reviewer #2: Yes

4. Is the manuscript presented in an intelligible fashion and written in standard English?

Reviewer #1: Yes

Reviewer #2: Yes

5. Review Comments to the Author

Reviewer #1: Manuscript Number: PONE-D-20-08484

Manuscript Title: The development and validation of a social media fatigue scale: From a cognitive-behavioral-emotional perspective

Journal: PLOS ONE

Athors: Shiyi Zhang, Yanni Shen, Tao Xin1, Haoqi Sun, Yilu Wang, Xiaotong Zhang, Siheng Ren

Subject: Review

Social media shorten the space distance between people and provide an online platform for them to make friends and to post information about themselves.Several surveys, however, showed that individuals tend to engage themselves less in social media in spite of the increasing popularity of social network services.Overall, some studies focus on understanding SMF and exploring its relationships with psychological determinations, however, to date few researches have used a standard psychological measurement of SMF. To fill this gap, the current study was aimed to develop and validate a measurement of SMF.

1. Title: The title is adequate.

2. Abstract: The abstract is NOT adequately addressed study, it should re-structured according to the journal format.

3. Introduction: The introduction is adequate.

4. Materials and Methods:

We recruited 30 participants offline for semi-structured interviews, and 1569 participants on SO JUMP (an online platform for data collection) in 2017. Data were entered using Epi-data software and transferred to SSPS version 21 for analysis. Data of 509 participants were used to explore the scale’s structure; dataof 552 participants were used to verify the scale’s structure and evaluate its properties; data of 508 participants were used to evaluate measuring properties of the scale

a. Study design: Adequately has been described

b .Sampling technique: It has NOT been adequately described

c. Recruitment of Subjects: It has NOT been Adequately described (Convenient sampling???)

d.Study duration: Provided,

e. Setting: Adequately has been described

f. Eligibility criteria; It has NOT been reported

g. Data collection and measurements: Data collection tool and analysis described in detail.

h. Ethical consideration: The study was approved by the ethics review committee of the Faculty of Psychology in Beijing Normal University.

5. Results: The present study showed main characteristics of SMF, which may also provide references for researchers, social media managers and operators to look for ways of mitigating people’s SMF. Firstly, the study verified that it was cognitive overload that served as one of the main characteristics of SMF. The more information people got accessed to on social media, the more likely they would feel fatigue.

6. Discussion: The discussion is well written and adequately addressed.

7. Contribution to the literature: The authors should provide key points and the contribution of current study to literature and what messages are provided with the present study?

8. Limitations: The author reported some of the key limitations of this study in detailed

9. Conclusions:

The authors reported that present study explored and verified the structure of SMF and the 15-item SMFS can be used to understand the phenomena of SMF for further study, which is NOT solid and conclusive at this stage.

Overall, this study addresses an important public health issue and well written. However, the methods section is grossly deficient from epidemiological and statistical point view The authors recruited 30 participants offline for semi-structured interviews, and 1569 participants on SO JUMP (an online platform for data collection) It seems 30 sıbjctes particioants are non-randomly selected and too small sample size, which can be considered as a pilot study. The authors should provide key points and the contribution of current study to literature and what messages are provided with the present study? The author reported some important limitations of this study in detailed. The current study addresses an important issues concerning public health and can be adopted in other countries. Although, the study does not contribute novel knowledge and solid conclusive results at this stage, but, it would help local policy makers.

Reviewer #2: The study was well designed and executed.

Classified with minor modifications, I recommend some changes and suggestions, they are: the exact repetition of writing should be avoided (see the sentence in the lines: 93 - 95 "when exposed to long-term excessive information, individuals with SMF will feel overloaded and eventually tend to refrain from using social media ", are also repeated on lines 100 - 101).

I also suggest a revision in the sentence (lines 269-272) "However, based on the literature review and structure assumption for SMF, the factor loadings pattern for the current four-factor model was nearly the assumed structure for SMF and it was thus chosen as the optimal SMFS structure. " As described, it seems that the four-factor model was chosen, which is not true.

Still in the results, in Table 5 it is recommended to include the factorial loads of the items in all factors, highlighting them in each of the three dimensions. And put the acronym for standard deviation in capital letters.

Regarding reliability it is also recommended to include the value of McDonald's Omega.

6. PLOS authors have the option to publish the peer review history of their article (what does this mean?). If published, this will include your full peer review and any attached files.

Reviewer #1: Yes: Prof. Dr. Abdulbari Bener

Reviewer #2: No

---

## [Author Response · Author response to Decision Letter 0]

11 Aug 2020

Dear editors and reviewers:

Thank you for your letter and the reviewers’ comments on our manuscript entitled The development and validation of a social media fatigue scale: From a cognitive-behavioral-emotional perspective (ID: PONE-D-20-08484). Those comments are very helpful for revising and improving our paper. We have studied the comments carefully and made corrections which we hope meet with approval. The main corrections are in the manuscript and the responds to the reviewers’ comments are as follows.

Replies to the reviewers’ comments:

Reviewer #1:

1. Abstract: The abstract is NOT adequately addressed study, it should re-structured according to the journal format.

Response: Thanks for your advice. We’ve re-structured a new abstract (line 22-45 in manuscript) according to your advice, and hope that it meets with approval.

2. Materials and methods:

Sampling technique: It has NOT been adequately described.

Recruitment of Subjects: It has NOT been Adequately described (Convenient sampling???)

Eligibility criteria; It has NOT been reported

Response: Thanks for your advice. We’ve added description of sampling technique (line 170-175 in manuscript), recruitment of subjects (line 170-175 in manuscript) and eligibility criteria (line 175-179 in manuscript). Hope that it will meet with approval. 

3. Contribution to the literature: The authors should provide key points and the contribution of current study to literature and what messages are provided with the present study?

Response: Thanks for your advice. In order to provide key points and the contribution current study, we’ve revised the discussion part, and added a summary paragraph (line 387, line 401, line 412, and line 426-439 in manuscript)

4. Conclusions:

The authors reported that present study explored and verified the structure of SMF and the 15-item SMFS can be used to understand the phenomena of SMF for further study, which is NOT solid and conclusive at this stage.

Response: Thanks for your advice. We’ve changed the expression in both abstract (line 39-45 in manuscript) and discussion (line 426-439 in manuscript). Hope it will meet with approval.

5. However, the methods section is grossly deficient from epidemiological and statistical point view The authors recruited 30 participants offline for semi-structured interviews, and 1569 participants on SO JUMP (an online platform for data collection) It seems 30 subjects participants are non-randomly selected and too small sample size, which can be considered as a pilot study.

Response: Thanks for your advice. We’ve considered the semi-structured interviews as a pilot study and revised the relative part in the paper (line 154, line 170-172, line 249-269 in manuscript), besides, we added information on sampling technique (line 170-175 in manuscript), recruitment of subjects (line 170-175 in manuscript) and eligibility criteria (line 175-179 in manuscript). Hope that it will make the methods section more clarify.

Reviewer #2:

1. I recommend some changes and suggestions, they are: the exact repetition of writing should be avoided (see the sentence in the lines: 93 - 95 "when exposed to long-term excessive information, individuals with SMF will feel overloaded and eventually tend to refrain from using social media ", are also repeated on lines 100 - 101).

Response: Thanks for your advice! We’ve rewritten the latter sentence which can be seen in line 107-108 of the manuscript “long term exposure to excessive information makes individuals with SMF feel overloaded and ultimately tend to reduce their use of social media”.

2. I also suggest a revision in the sentence (lines 269-272) "However, based on the literature review and structure assumption for SMF, the factor loadings pattern for the current four-factor model was nearly the assumed structure for SMF and it was thus chosen as the optimal SMFS structure. " As described, it seems that the four-factor model was chosen, which is not true.

Response: Thank you for your advice, and we apologize for our carelessness! We’ve changed “four-factor model” into “three-factor model” (line 290 in the manuscript).

3. Still in the results, in Table 5 it is recommended to include the factorial loads of the items in all factors, highlighting them in each of the three dimensions. And put the acronym for standard deviation in capital letters.

Response: Thanks for your advice! We included the factorial loads of the items in all factors in Table 5, and we also put the acronym for standard deviation in capital letters.

4. Regarding reliability it is also recommended to include the value of McDonald's Omega.

Response: Thanks for your advice. We’ve calculated the McDonald’s Omega of the data, and replaced all the data of Cronbach’s alpha with McDonald’s Omega. 

Once again, thank you very much for your constructive comments and suggestions which would help us both in English and in depth to improve the quality of the paper.

Kind regards,

Tao Xin

E-mail: xtao_bnu@163.com

---

## [Decision Letter · Decision Letter 1]

14 Sep 2020

PONE-D-20-08484R1

The development and validation of a social media fatigue scale: From a cognitive-behavioral-emotional perspective

PLOS ONE

Dear Dr. Xin,

Thank you for submitting your manuscript to PLOS ONE. After careful consideration, we feel that it has merit but does not fully meet PLOS ONE’s publication criteria as it currently stands. Therefore, we invite you to submit a revised version of the manuscript that addresses the points raised during the review process.

It is fundamental to address the comments made by reviewer 1 on the first and second review.

We look forward to receiving your revised manuscript.

Kind regards,

Geilson Lima Santana, M.D., Ph.D.

Academic Editor

PLOS ONE

Reviewers' comments:

Reviewer's Responses to Questions

**Comments to the Author**

1. If the authors have adequately addressed your comments raised in a previous round of review and you feel that this manuscript is now acceptable for publication, you may indicate that here to bypass the “Comments to the Author” section, enter your conflict of interest statement in the “Confidential to Editor” section, and submit your "Accept" recommendation.

Reviewer #1: (No Response)

Reviewer #2: All comments have been addressed

2. Is the manuscript technically sound, and do the data support the conclusions?

Reviewer #1: Partly

Reviewer #2: Yes

3. Has the statistical analysis been performed appropriately and rigorously? 

Reviewer #1: No

Reviewer #2: Yes

4. Have the authors made all data underlying the findings in their manuscript fully available?

Reviewer #1: Yes

Reviewer #2: Yes

5. Is the manuscript presented in an intelligible fashion and written in standard English?

Reviewer #1: No

Reviewer #2: Yes

6. Review Comments to the Author

Reviewer #1: Overall, this study addresses an important public health issue, but the manuscript is not written as clear and straightforward. It seems subjects or participants are non-randomly selected which subject and conclusion might be considered as a bias. Also, the manuscript has been presented such as report; there are so many unnecessary heading and subtitles. The manuscript, unnecessary has been expended and reader can be feeling boring. It could be written in very professional, clear and concise way. Although, the study does not contribute novel knowledge or add sufficiently to the current literature, but, it would help local policy makers. I think the major concern of this submission is it lacks sufficient novelty and or original study.

Reviewer #2: All the recommendations suggested were implemented and the manuscript it is more clear and standardized according to Plos One guidelines. In this way, the article now has more robustness and looks good for publication in the journal.

7. PLOS authors have the option to publish the peer review history of their article (what does this mean?). If published, this will include your full peer review and any attached files.

Reviewer #1: No

Reviewer #2: **Yes: **Tailson Mariano

---

## [Author Response · Author response to Decision Letter 1]

20 Nov 2020

Dear editors and reviewers:

Thank you for your letter and the reviewers’ comments on our manuscript entitled The development and validation of a social media fatigue scale: From a cognitive-behavioral-emotional perspective (ID: PONE-D-20-08484). Those comments are very helpful for revising and improving our paper. We have studied the comments carefully and made corrections which we hope will meet with approval. The main corrections are in the manuscript and the responds to the reviewers’ comments are as follows.

Replies to the reviewers’ comments:

Reviewer #1:

Comments and its replies on the first review:

1. Abstract: The abstract is NOT adequately addressed study, it should re-structured according to the journal format.

Response:

Thanks for your advice. We’ve re-structured a new abstract (line 22-45 in manuscript) according to your advice, and hope that it will meet with approval.

2. Materials and methods:

Sampling technique: It has NOT been adequately described.

Recruitment of Subjects: It has NOT been Adequately described (Convenient sampling???)

Eligibility criteria; It has NOT been reported

Response: 

Thanks for your advice. We’ve added description of sampling technique (line 168-173 in manuscript), recruitment of subjects (line 168-173 in manuscript) and eligibility criteria (line 173-177 in manuscript). Hope that it will meet with approval. 

Let us give you some brief information here:

- Sampling technique: As for the pilot study, convenient samples (30 university students in Beijing) were utilized. As for other studies, simple random sampling was conducted (1569 participants from different places in mainland China).

- Recruitment of subjects: As for the pilot study, participants were recruited through recruitment advertisement. As for other studies, participants were recruited online through a platform where participants at various ages come from different places in mainland China. 

- Eligibility criteria: The inclusion criterion was that every participant had used social media within one month before he/she took part in the study. Besides, according to response time and other factors, the platform could automatically eliminate those participants who didn’t complete the scale seriously.

3. Contribution to the literature: The authors should provide key points and the contribution of current study to literature and what messages are provided with the present study?

Response: 

Thanks for your advice. In order to provide key points and the contribution current study, we’ve revised the discussion part, and added a summary paragraph (line 371, line 385, line 396, and line 410-423 in manuscript).

4. Conclusions:

The authors reported that present study explored and verified the structure of SMF and the 15-item SMFS can be used to understand the phenomena of SMF for further study, which is NOT solid and conclusive at this stage.

Response: 

Thanks for your advice. We’ve changed the expression in both abstract (line 39-45 in manuscript) and discussion (line 410-423 in manuscript). Hope it will meet with approval.

5. However, the methods section is grossly deficient from epidemiological and statistical point view The authors recruited 30 participants offline for semi-structured interviews, and 1569 participants on SO JUMP (an online platform for data collection) It seems 30 subjects participants are non-randomly selected and too small sample size, which can be considered as a pilot study.

Response: 

Thanks for your advice. We’ve considered the semi-structured interviews as a pilot study and revised the relative part in the paper (line 153-158, line 168-170, line 233-253 in manuscript), besides, we added information on sampling technique (line 168-173 in manuscript), recruitment of subjects (line 168-173 in manuscript) and eligibility criteria (line 173-177 in manuscript). Hope that it will make the methods section more clarify.

Comments and its replies on the second review:

1. If the authors have adequately addressed your comments raised in a previous round of review and you feel that this manuscript is now acceptable for publication, you may indicate that here to bypass the “Comments to the Author” section, enter your conflict of interest statement in the “Confidential to Editor” section, and submit your "Accept" recommendation.

Reviewer #1: (No Response)

Response: 

It's a pity that the changes in the previous edition have not completely meet with approval. This time, we revised both our replies and manuscript in further, hoping that our replies and revision could address your comments in a more adequate way. (The specific content has been described in each corresponding comment.)

2. Is the manuscript technically sound, and do the data support the conclusions?

Reviewer #1: Partly

Response: 

We’re sorry that our previous revision may not be strictly standardized and concise. According to your suggestion, we have improved the article, and the specific improvement could be seen in the coming part. We hope that our revision can be more standardized and professional than the previous version.

3. Has the statistical analysis been performed appropriately and rigorously? 

Reviewer #1: No

Response:

We’re sorry that our previous revision may not be rigorous enough in data analysis. In the new revision, we improved the statement on sampling and collected a small group of data to test whether the results of convenient samples were biased. (line 425-437 in manuscript)

4. Have the authors made all data underlying the findings in their manuscript fully available?

Reviewer #1: Yes

Response:

Thank you for your recognition!

5. Is the manuscript presented in an intelligible fashion and written in standard English?

Reviewer #1: No

Response:

Thank you for your advice! The previous revision was not presented clear enough, or the title may be confusing, we’re sorry for that! We’ve made improvements according to your suggestion. As a result, the present revision only has the first- and second- level heading. Hope that this will make the structure of the article clearer.

6. Review Comments to the Author

Reviewer #1: 

Overall, this study addresses an important public health issue, but the manuscript is not written as clear and straightforward. It seems subjects or participants are non-randomly selected which subject and conclusion might be considered as a bias. Also, the manuscript has been presented such as report; there are so many unnecessary heading and subtitles. The manuscript, unnecessary has been expended and reader can be feeling boring. It could be written in very professional, clear and concise way. Although, the study does not contribute novel knowledge or add sufficiently to the current literature, but, it would help local policy makers. I think the major concern of this submission is it lacks sufficient novelty and or original study.

Response:

Thanks for your recognition and advice! We’ll reply to your comments from the following aspects:

1. “It seems subjects or participants are non-randomly selected which subject and conclusion might be considered as a bias.”: 

It is true that participants in pilot study were convenient samples. We did it for the consideration that after being interviewed, these participants would be re-invited to complete the raw scale and offer their suggestions to improve it. To ensure that the pilot study would go on wheels, we decided to use convenient samples. 

Thanks for your question, we collected another group of participants (random sample) to test whether the convenient sample will lead to bias in our study, and the result indicated that bias may not exist here (line 425-437 in manuscript).

2. “the manuscript is not written as clear and straightforward”, “The manuscript, unnecessary has been expended and reader can be feeling boring. It could be written in very professional, clear and concise way.”, “the manuscript has been presented such as report; there are so many unnecessary heading and subtitles”:

Thanks for your comments! In order to revise our article, we refer to an article named Development and validation of the Scale of Motives for Using Social Networking Sites (SMU-SNS) for adolescents and youths (from Plos One, https://doi.org/10.1371/journal.pone.0225781), and did some revision. Furthermore, according to your advice, we deleted the unnecessary heading and subtitles, and we also simplify some expressions in this article.

In addition, the new file named Revised Manuscript with Track Changes only contains the modified track of content change (the track with format changes was deleted). We hope that it will make the manuscript look clearer.

3. “it lacks sufficient novelty and or original study.”:

The method of the study may not novel enough, but it may have its own contribution both in theoretical and practical aspects:

1) It clarified the three-factor structure of SMF, breaking through the unidimensional perspective.

2) It expanded the utilization of LCM to interpret SMF, from only one perspective (cognitive aspect) to three perspectives (cognitive, behavioral and emotional aspects).

3) It has developed an SMFS in a relative reliable way, and makes up for the lacks of previous studies which only used few items (usually, those items varied in different studies) to assess SMF without detailed reports of their psychometric properties.

4) It showed main characteristics of SMF, which may also provide references for researchers, social media managers and operators to look for ways of mitigating people’s SMF.

Reviewer #2:

Comments and its replies on the first review:

1. I recommend some changes and suggestions, they are: the exact repetition of writing should be avoided (see the sentence in the lines: 93 - 95 "when exposed to long-term excessive information, individuals with SMF will feel overloaded and eventually tend to refrain from using social media ", are also repeated on lines 100 - 101).

Response: 

Thanks for your advice! We’ve rewritten the latter sentence which can be seen in line 106-107 of the manuscript “long term exposure to excessive information makes individuals with SMF feel overloaded and ultimately tend to reduce their use of social media”.

2. I also suggest a revision in the sentence (lines 269-272) "However, based on the literature review and structure assumption for SMF, the factor loadings pattern for the current four-factor model was nearly the assumed structure for SMF and it was thus chosen as the optimal SMFS structure. " As described, it seems that the four-factor model was chosen, which is not true.

Response: 

Thank you for your advice, and we apologize for our carelessness! We’ve changed “four-factor model” into “three-factor model” (line 274 in the manuscript).

3. Still in the results, in Table 5 it is recommended to include the factorial loads of the items in all factors, highlighting them in each of the three dimensions. And put the acronym for standard deviation in capital letters.

Response: 

Thanks for your advice! We included the factorial loads of the items in all factors in Table 5, and we also put the acronym for standard deviation in capital letters.

4. Regarding reliability it is also recommended to include the value of McDonald's Omega.

Response: 

Thanks for your advice. We’ve calculated the McDonald’s Omega of the data, and replaced all the data of Cronbach’s alpha with McDonald’s Omega. 

Comments and its replies on the second review:

All the recommendations suggested were implemented and the manuscript it is more clear and standardized according to Plos One guidelines. In this way, the article now has more robustness and looks good for publication in the journal.

Response: 

Thanks for your recognition. It was the reviewers’ comments and advice that helped us to make the article clearer and more standardized!

Once again, thank you very much for your constructive comments and suggestions which would help us both in English and in depth to improve the quality of the paper.

Kind regards,

Tao Xin

E-mail: xtao_bnu@163.com

---

## [Decision Letter · Decision Letter 2]

2 Jan 2021

The development and validation of a social media fatigue scale: From a cognitive-behavioral-emotional perspective

PONE-D-20-08484R2

Dear Dr. Xin,

We’re pleased to inform you that your manuscript has been judged scientifically suitable for publication and will be formally accepted for publication once it meets all outstanding technical requirements.

Kind regards,

Geilson Lima Santana, M.D., Ph.D.

Academic Editor

PLOS ONE

Additional Editor Comments (optional):

Reviewers' comments:

Reviewer's Responses to Questions

**Comments to the Author**

1. If the authors have adequately addressed your comments raised in a previous round of review and you feel that this manuscript is now acceptable for publication, you may indicate that here to bypass the “Comments to the Author” section, enter your conflict of interest statement in the “Confidential to Editor” section, and submit your "Accept" recommendation.

Reviewer #1: All comments have been addressed

Reviewer #2: All comments have been addressed

2. Is the manuscript technically sound, and do the data support the conclusions?

Reviewer #1: Yes

Reviewer #2: Yes

3. Has the statistical analysis been performed appropriately and rigorously? 

Reviewer #1: Yes

Reviewer #2: Yes

4. Have the authors made all data underlying the findings in their manuscript fully available?

Reviewer #1: Yes

Reviewer #2: Yes

5. Is the manuscript presented in an intelligible fashion and written in standard English?

Reviewer #1: Yes

Reviewer #2: Yes

6. Review Comments to the Author

Reviewer #1: I am very pleased to confirm that the authors have satisfactorily incorporated all the suggested changes and the revised manuscript, although, I think the major concern of this submission is it lacks sufficient novelty and or original study, as many of such similar studies were seen published in currently documented literature.

Reviewer #2: All my previous suggestions were addressed, then once again I recommend the acceptance of the manuscript. In my opinion, this study was performed following a rigorous psychometric method and has its valor for publication. After all the reviews, the text is now more clear and technical. Similarly, the conclusions and limitations are now well explained and delimited.

Develop a psychological measure is a long hard path and do it right is yet harder. This study does it well and contributes to science with a psychometric trustful measure.

7. PLOS authors have the option to publish the peer review history of their article (what does this mean?). If published, this will include your full peer review and any attached files.

Reviewer #1: No

Reviewer #2: **Yes: **Tailson Mariano, PhD in Social Psychology

---

## [Editor Report · Acceptance letter]

11 Jan 2021

PONE-D-20-08484R2 

The development and validation of a social media fatigue scale: From a cognitive-behavioral-emotional perspective 

Dear Dr. Xin:

I'm pleased to inform you that your manuscript has been deemed suitable for publication in PLOS ONE. Congratulations! Your manuscript is now with our production department. 

Kind regards, 

on behalf of

Dr. Geilson Lima Santana 

Academic Editor

PLOS ONE